

# The antimicrobial activity of tea tree oil (*Melaleuca alternifolia*) and its metal nanoparticles in oral bacteria

Afrah E. Mohammed[1], Reham M. Aldahasi[1], Ishrat Rahman[2], Ashwag Shami[1], Modhi Alotaibi[1], Munerah S. BinShabaib[3], Shatha S. ALHarthi[3] and Kawther Aabed[1]

[1] Department of Biology, College of Science, Princess Nourah bint Abdulrahman University, Riyadh, Saudi Arabia
[2] Department of Basic Dental Sciences, College of Dentistry, Princess Nourah bint Abdulrahman University, Riyadh, Saudi Arabia
[3] Department of Preventive Dental Sciences, College of Dentistry, Princess Nourah bint Abdulrahman University, Riyadh, Saudi Arabia

Corresponding author
Kawther Aabed, Kfabed@pnu.edu.sa

## ABSTRACT

Tea tree (*Melaleuca alternifolia*) oil (TTO) is an antimicrobial agent, and hence, its use in fabricating nanoparticles (NP) may be useful in providing more efficacious antimicrobial agents. The current research aimed to test the antimicrobial efficacy of TTO and its TTO-Metal-NPs against oral microbes: *Porphyromonas gingivalis*, *Enterococcus faecalis*, and *Streptococcus mutans*. The antimicrobial activity of TTO and zinc (Zn) and iron (Fe) nanoparticles (NPs) and the combined effects of antimicrobial agents were investigated using agar well diffusion assays. Fourier-transform infrared spectroscopy (FT-IR) was used to identify the phyto-constituents of TTO. Field emission scanning electron microscopy (FE-SEM), dynamic light scatter (DLS), and zeta potential were utilized to analyze the biogenic nanoparticles' morphology, size, and potential. The antimicrobial mode of action was determined by assessing the morphological changes under scanning electron microscopy (SEM). The TTO extracts converted Zn and Fe ions to NPs, having an average size of 97.50 (ZnNPs) and 102.4 nm (FeNPs). All tested agents had significant antibacterial efficacy against the tested oral microbes. However, the TTO extract was more efficacious than the NPs. Combination treatment of TTO with antibiotics resulted in partial additive effects against *P. gingivalis* and partial antagonistic effects against *E. faecalis*, *S. mutans*, and common mouthwashes (Oral B and chlorhexidine). TTO and NP-treated bacteria underwent morphological changes on treatment. *M. alternifolia* phytochemicals could be useful for further research and development of antimicrobial NPs. The current study highlights the variance in activity observed for different types of bacteria and antagonistic effects seen with common mouthwashes, which represent a threat to therapeutic efficacy and heighten the risk of clinical microbial resistance.

## INTRODUCTION

Microorganisms mostly enter the whole gastrointestinal system through the mouth (*Manji, Dahlen & Fejerskov, 2018*), acting as an interface and a port for their colonization and growth (*Gedif Meseret, 2021*). Plaque is a pathogenic biofilm made up of host bacteria having complex interactions, causing oral illnesses such as dental caries and periodontal diseases. Several acidogenic and acidic bacteria have been discovered in dental plaque that initiate and develop dental caries (*Panpaliya et al., 2019*). A buildup of plaque can also provoke gingivitis, a reversible form of periodontal disease. Periodontitis is characterized by the development of the periodontal pocket and bone recession, causing the degeneration of the tooth support system (*Fiorillo et al., 2019*), leading to tooth mobility and loss. In the past 20 years, it has been suggested that periodontitis might be a risk factor for various systemic disorders, including cardiovascular conditions (*Mei et al., 2020*).

Numerous bacterial species are present in the oral cavity, each attributed to a specific oral condition. *Porphyromonas gingivalis* (*P. gingivalis*) is a pathogenic rod-shaped, Gram-negative anaerobic bacterium growing in colonies on dark blood agar (*Xu et al., 2020*). It is associated with the development of periodontitis and proved to be closely correlated with the occurrence and development of many systemic diseases, such as atherosclerosis, cancer, and Alzheimer's disease (*Dominy et al., 2019*; *Wang et al., 2021*; *Zhang et al., 2021*). On the other hand, *streptococcus mutans* is a facultative anaerobe Gram-positive cocci (*Ng et al., 2020*) that colonizes the dental surface and damages the hard tooth structure in the presence of fermentable carbohydrates. *S. mutan* s can bind to the enamel salivary pellicle and other plaque bacteria, resulting in increased dental caries (*Salehi et al., 2020*). It is the most common infection associated with dental decay and is the most prevalent oral infection (*Aqawi et al., 2021*). *Enterococcus faecalis* is also a Gram-positive facultative anaerobic bacteria widely found in the human oral cavity and gastrointestinal system (*Deng et al., 2023*). It may also cause various diseases that impact the body's overall health, such as endocarditis, bacteremia, urinary tract infections, meningitis, and root canal infections (*Bolocan et al., 2019*).

The emergence of antimicrobial resistance is a significant public health concern. Antibiotics, whether natural or synthetic, are chemical compounds that effectively combat microbes (*Al Alsheikh et al., 2020*) which operate as either bacteriocidal agents or an inhibitor of microbial growth (bacteriostatic agents). Both types are used regularly to treat bacterial infections in people and animals (*Serwecińska, 2020*), including infections in the oral cavity (*Zannella et al., 2020*). Public health is at stake due to the rise in pathogenic infections and antibiotic-resistant bacteria. The mortality and morbidity rates linked to this risk are higher than those linked to breast, prostate, and HIV cancers (*Serwecińska, 2020*). In addition, with growing antibiotic consumption and the rise of opportunistic microorganisms naturally resistant to drugs, current treatment regimens become ineffective, and diseases become more difficult to treat (*Al Alsheikh et al., 2020*). Therefore, developing novel, safe, cost-effective, and efficacious antimicrobial agents is a priority. Furthermore, metal ions may be effective as antibacterial agents, where their chemical and physical characteristics enhance their cytotoxic properties (*Godoy-Gallardo*

*et al., 2021*). Iron and zinc (Zn) ions have received the most attention because of their ubiquitous natural abundance and frequent daily use (*Ye et al., 2020*). Iron is one of the most widespread elements on Earth, ranking fourth in the Earth's crust (*Gudkov et al., 2021*). It is also a vital microelement that plays a significant part in the function of biological systems (*Gudkov et al., 2021*). Zinc is a crucial trace element, abundant in the environment and the human body, and, in large amounts, prevents bacterial development. On the other hand, metals in the form of nanoparticles (NPs) have grown in appeal as a therapy for resistant diseases due to their potent antibacterial properties. As a result, it is gaining popularity as a potential antibiotic alternative (*Zannella et al., 2020*). Nanotechnology is a rapidly expanding science that creates new nano-scale materials (*Pillai et al., 2020*). Nanomaterials are a promising option for fighting bacteria due to their distinct physicochemical properties attributed to their small size and large surface area (*Chatzimitakos & Stalikas, 2016*).

Metal NPs have demonstrated broad-spectrum antibacterial efficacy against Gram positive and Gram negative bacteria by modifying the metabolic activity of bacteria (*Chatzimitakos & Stalikas, 2016*). Iron oxide nanoparticles (FeONPs) are incredibly important and have recently gained attention due to their core properties and unusual physiochemical characteristics (*Salgado et al., 2019*). Moreover, FeNPs have proven to be effective against various pathogenic bacterial strains and fungi due to their ability to generate highly reactive oxygen species (ROS) (*Muthukumar et al., 2018*). More recently, zinc oxide nanoparticles (ZnONPs) have gained the interest of academics and scientists as effective antimicrobial agents due to their wide range of medicinal applications (*Jadoun et al., 2021*).

One of the best and safest ways to synthesize nanoparticles is by using biological entities such as microorganisms or plant extracts (*Bahrulolum et al., 2021*). Plant extracts are commonly employed because they are readily available, and there are no restrictions on their use in microbe applications (*Mohammed & Al-Megrin, 2021*).

Plants include a variety of metabolites and biochemicals (for example, polyphenols) that can act as both a stabilizing and reducing agent in the production of biogenic NPs (*Singh et al., 2020*). Previous research synthesized iron and zinc nanoparticles from various plant extracts and investigated their antibacterial effects (*Theophil Anand et al., 2019*; *Vitta et al., 2020*; *Álvarez-Chimal et al., 2021*; *Al-Karagoly et al., 2022*). In the current study, *Melaleuca alternifolia*, tea tree oil (TTO) extract was used as a biogenic agent for FeNPs and ZnNPs fabrication. TTO is extracted from a small tree of the Myrtaceae family endemic to Australia (*Borotová et al., 2022*). TTO is widely accessible in markets worldwide (*Mohammed et al., 2023a*) with broad-spectrum antimicrobial activities, frequently used to treat infections caused by fungi, bacteria, and viruses (*Qi et al., 2021*). Terpinen-4-ol and -terpineol are important antibacterial metabolites in tea tree oil that have antibacterial abilities against *E. faecalis* and *P. fluorescens* (*Borotová et al., 2022*). In addition, TTO has been shown to reduce oral bacteria in root canals and inhibit *E. Faecalis* (*Qi et al., 2021*) and *P. gingivalis* (*Mohammed et al., 2023a*).

Our research aims to explore an eco-friendly approach to Zn and Fe nanoparticle fabrication using TTO. The physicochemical properties of the NPs were evaluated using field emission scanning electron microscopy techniques, dynamic light scatter, and zeta

potential besides fourier-transform infrared spectroscopy. Furthermore, TTO extract, ZnNPs, and FeNPs were tested against some oral pathogenic bacteria, *P. gingivalis*, *E. faecalis*, and *S. mutans*, and morphological changes for treated bacteria were determined using SEM analysis.

## MATERIAL AND METHODS

### Sample collection and preparation
Tea tree oil (*Melaleuca alternifolia*) was bought from a local market in Riyadh, Saudi Arabia. Aqueous extracts were prepared by adding 2 ml of TTO to 100 ml of distilled water. Heat treatment for 15 min at 80 °C was performed, and then the mixture was filtered through Whatman candidate No. 1 (pore size 125 mm, Whatman, Maidstone, UK). Filtrates were kept at 4 °C for subsequent use.

### Nanoparticles fabrication
Zinc acetate dihydrate was prepared at a concentration of 0.01 M, and ferric and ferrous chloride hexahydrate was prepared individually at concentrations of 0.01M and then mixed at a ratio of 2:1 (*Mohammed, Algebaly & Elobeid, 2021*) for NP fabrication. TTO (10 mL, 2%) was used as a reducing and capping agent, mixed with 90 ml of individual metal salt solution in an Erlenmeyer flask, and then allowed to react at room temperature for 48 h. A change in the color of the mixtures indicated the formation of nanoparticles. NPs were separated from the mixtures by centrifugation at 12,000 rpm for 15 min. Then, the NP pellet was washed with distilled water, and the supernatents were discarded. Finally, the pellets were dried at room temperature and stored for further study at 4 °C (*Ahsan et al., 2020*).

### Characterization of nanoparticles
A Zetasizer nano device (Malvern, Worcestershire, UK) was used to measure the size and potential of NPs prepared by TTO. FT-IR measured the infrared absorption and emission spectra of the prepared samples. A range of 500–4,000 cm$^{-1}$ FT-IR (Nicolet 6700 FT-IR Spectrometer, Waltham, MA, USA) was used. Scanning Electron Microscopy (SEM) technique revealed information about the sample, including the external morphology of the material making up the samples. Treated bacteria were also visualized using SEM, and all images were taken at an accelerating voltage of 30 kV. JEOL JEM-2100 (JEOL, Peabody, MA, USA).

### Identification of TTO phytochemicals by GC-MS analysis
Gas chromatography-mass spectrometry (GC-MS) was used to analyze 50 µL of tea tree oil in one mL of methanol solvent. The analysis was conducted using Agilent Technologies 7890B Gas Detector and Agilent Technologies 7000D Mass Spectrometer Detector in Santa Clara, CA, USA. The analysis was carried out using helium (high purity) at 70 eV ionization energy and scanning the m/z range of 30–500. The transfer, source, and quadrupole temperatures were set at 245 °C, 230 °C, and 150 °C, respectively. An Agilent HP-5MS 5% phenyl methyl siloxane capillary column (30 m × 0.25 mm × 0.25 µm film thickness)

was used, which was set from 60 °C to 325 °C. Initially, the oven temperature was kept at 60 °C for 1.5 min. Then, it was raised to 160 °C in 3.5 min. Finally, the temperature was increased to 290 °C for 20 min to complete the experiment. In splitless mode, the injection volume was 10 μl, and the injector temperature was 280 °C.

### Evaluation of the antibacterial activity of the NPs

The antibacterial activity of TTO (2%) and NPs (1 mg/mL) was determined using agar-well diffusion methods (*Lakkim et al., 2020*). Three types of bacteria, *P. gingivalis* from the Gram-negative group, E. faecalis, and S. mutans from the Gram-positive group, were tested. Pure cultures of the microorganisms were sub-cultured on Mueller-Hinton Agar, and then 0.2 ml of bacterial strain ($1.5 \times 10^8$ CFU/ml) was swabbed (sterile swabs) uniformly onto individual agar plates. Subsequently, three adequately spaced wells (holes), each four mm in diameter, were made per plate at the culture agar surface using a sterile metal cork borer. In each hole, 0.2 mL of tested agents were added individually under aseptic conditions and kept at room temperature for one hour to allow the agents to diffuse into the agar medium. Finally, the plates were incubated at 37 °C for 18–24 h. Sterile distilled water was used as the negative reference control. Inhibition zones that appeared as clear areas around the wells were evaluated.

### Minimum inhibitory concentration and minimum bactericidal concentration

The minimum inhibitory concentration (MIC) and minimum bactericidal concentration (MBC) values were determined by a microdilution method in nutrient broth (NB). one mL of a bacterial strain containing $1.5 \times 10^8$ CFU/ml of bacteria was added individually to 10 ml of NB. Different concentrations of NPs were added to the test tubes containing the bacterial strains and incubated for 24 h. After incubation, the MIC values were obtained by checking the tube turbidity as an indication of bacterial growth. The MIC value corresponded to the concentration that inhibited 99% of bacterial growth, and the MBC was the concentration that killed 99.9% of bacterial growth (*Sonbol, Mohammed & Korany, 2022*).

### Antibacterial potential of tea tree oil, antibiotics, and their combinations

The antibacterial effects of the tea tree oil (2%), oral B, and chlorhexidine (0.2%), as well as standard antibiotics, Flagyin (500 mg), and Amoxicillin (500 mg) were tested against *P. gingivalis*, *E. faecalis*, and *S. mutans* using the disk diffusion method (*Larsen, 2002*; *Binshabaib et al., 2020*; *Qi et al., 2021*), respectively. For the synergistic antibacterial activity test, freshly cultured bacteria were used, and one mL of TTO (2%) was added to the antibiotic disks and one mL of mouthwash, individually. The discs were placed on the plates cultured with bacteria, and the mixed solutions were added to the prepared wells, as described above. The synergistic antibacterial activity of tea tree oil and the combination was measured after 24 h of incubation at 37 °C as the diameter of the inhibition zone around the disks and wells (mm), respectively.

## Statistical analysis

Each test in the present investigation has been repeated a minimum of three times; statistical analysis and figures were prepared using GraphPad Prism Software version 9.1 (San Diego, CA, USA). The SEM images of the NPs were chosen as a representative from one of the repetitions.

## RESULTS AND DISCUSSION

Evaluating extracts from plant origin as natural antimicrobial agents against microbes is needed due to the development of antibiotic resistance. Hereby, TTO was used alone and in combination with antibiotics to determine any useful synergistic activity against oral pathogens. In addition, combination activity was tested with fabricated biogenic Zn and FeNPs.

### NPs synthesis using TTO and their characterization

The application of nanomaterials has grown in recent years attributed to the properties exhibited due to the high surface-volume ratio differing from their bulk materials counterparts (*Yaqoob et al., 2020*). The current study was designed to examine the efficacy of TTO as a biogenic agent in the fabrication of NPs and to detect their activities against some oral microbes. TTO successfully operated as a reducing agent for forming FeNPs and ZnNPs. The reaction substrate gradually changed from clear solution to turbid in a time-dependent form after TTO was mixed with each metal salt. Such a conversion was considered the primary sign for the biotransformation of $Fe^+$ ion to $Fe^0$ and $Zn^+$ ion to $Zn^o$ with the aid of TTO. Additional confirmation for NP formation was recorded by DLS (Figs. 1 and 2), showing an average size of 97.50 nm and 102.4 nm besides zeta potential of $-4.16$ and $-2.52$ for ZnNPs and FeNPs, respectively. Other studies also used essential oils as biogenic reducing agents in AgNP synthesis, which resulted in NPs at a range of 55 nm (*Vinicius de Oliveira Brisola Maciel et al., 2020*). Furthermore, ZnONPs (40 nm) and FeNPs (9.3–27 nm) were fabricated with the aid of essential oils from *Eucalyptus globulus* and *Satureja hortensis,* respectively (*Obeizi et al., 2020*; *Ahmadi et al., 2021*). Such NPs displayed significant antimicrobial activity and hence could be an effective substitute for antibiotics. In the current study, a negative zeta potential indicated the repulsion among the particles and a high degree of stability (*Bélteky et al., 2019*).

SEM images for both NPs prepared using TTO are presented in Figs. 3 and 4 showing spherical shapes with some particles aggregated. Using SEM analysis the diameter of prepared FeNPs and ZnNPs were found to be 30 nm and 20 nm, respectively. The differences between NP sizes detected by DLS and SEM techniques could be related to the fundamental principles of measurement for each instrument where DLS mainly determines the hydrodynamic radius of the NPs (*Khandel et al., 2018*). Supporting the current findings, a similar method of synthesizing green biogenic ZnONPs and FeONPs was used with *Dysphania ambrosioides* and *Nigella sativa* seed extract, respectively (*Álvarez-Chimal et al., 2021*; *Al-Karagoly et al., 2022*).

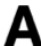

**A**

| | Size (d.nm): | % Intensity: | St Dev (d.n... |
|---|---|---|---|
| **Z-Average (d.nm):** 97.50 | **Peak 1:** 164.3 | 92.2 | 93.95 |
| **PdI:** 0.431 | **Peak 2:** 18.85 | 7.8 | 5.455 |
| **Intercept:** 0.964 | **Peak 3:** 0.000 | 0.0 | 0.000 |
| **Result quality :** Good | | | |

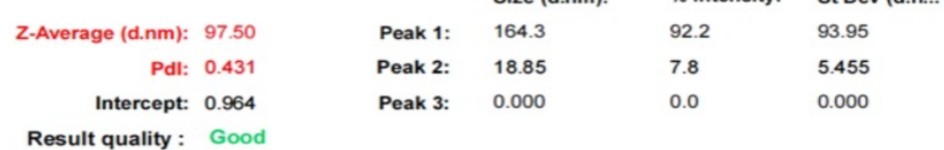

Record 323: Tf 1    Record 324: Tf 2    Record 325: Tf 3

**B**

| | Mean (mV) | Area (%) | St Dev (mV) |
|---|---|---|---|
| **Zeta Potential (mV):** -4.16 | **Peak 1:** -4.16 | 100.0 | 4.16 |
| **Zeta Deviation (mV):** 4.16 | **Peak 2:** 0.00 | 0.0 | 0.00 |
| **Conductivity (mS/cm):** 0.0312 | **Peak 3:** 0.00 | 0.0 | 0.00 |
| **Result quality :** Good | | | |

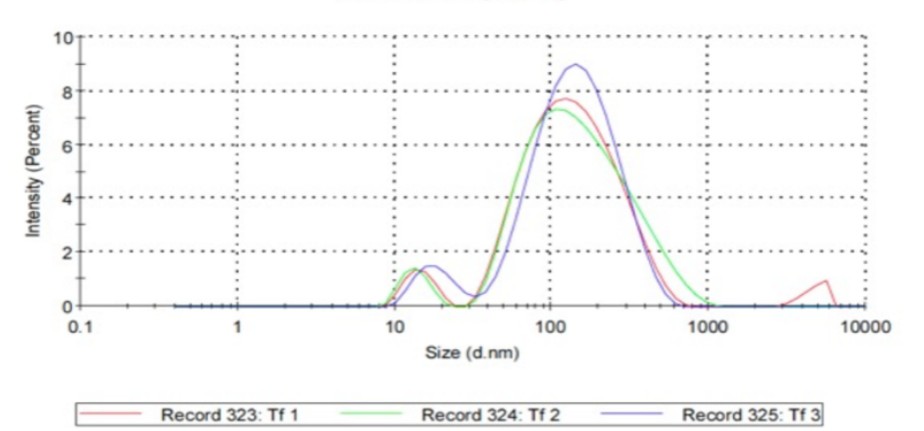

Record 320: Tf 1    Record 321: Tf 2    Record 322: Tf 3

**Figure 1** **Physiochemical properties of *M. alternifolia* (TTO) fabricated ZnNPs.** Zeta size distribution (A) and potential (B).

**A**

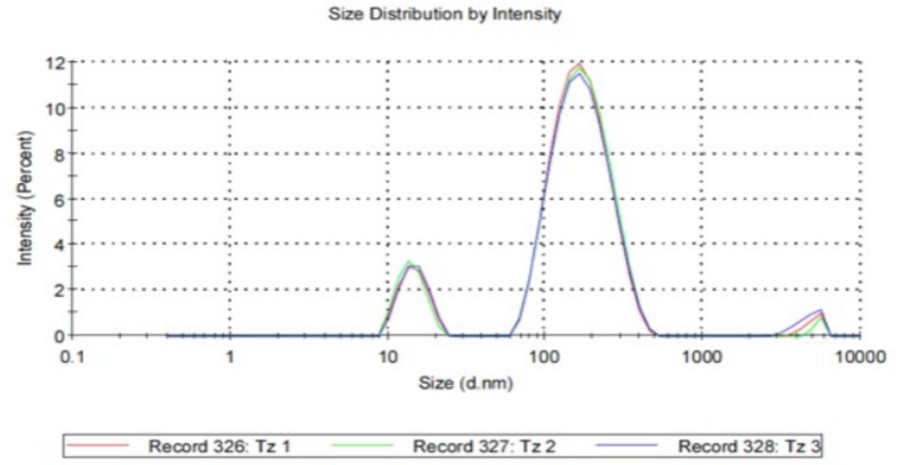

|  | | Size (d.nm): | % Intensity: | St Dev (d.n... |
|---|---|---|---|---|
| Z-Average (d.nm): 102.4 | Peak 1: | 179.1 | 84.6 | 72.73 |
| Pdl: 0.582 | Peak 2: | 14.94 | 12.0 | 2.947 |
| Intercept: 0.957 | Peak 3: | 4694 | 3.4 | 782.8 |
| Result quality : Refer to quality report | | | | |

**B**

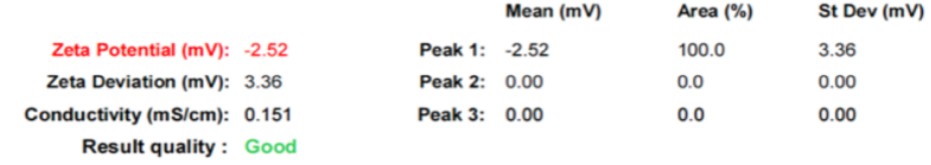

|  | | Mean (mV) | Area (%) | St Dev (mV) |
|---|---|---|---|---|
| Zeta Potential (mV): -2.52 | Peak 1: | -2.52 | 100.0 | 3.36 |
| Zeta Deviation (mV): 3.36 | Peak 2: | 0.00 | 0.0 | 0.00 |
| Conductivity (mS/cm): 0.151 | Peak 3: | 0.00 | 0.0 | 0.00 |
| Result quality : Good | | | | |

**Figure 2 Physiochemical properties of *M. alternifolia* (TTO) fabricated FeNPs.** Zeta size distribution (A) and potential (B).

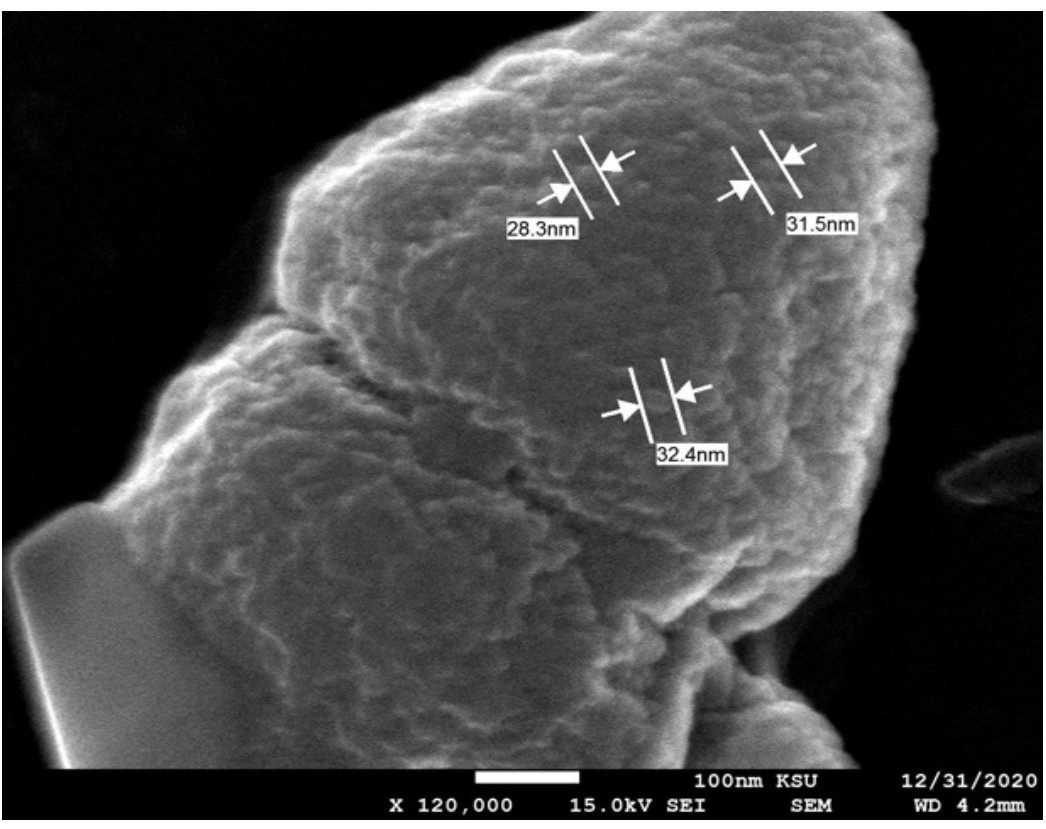

**Figure 3** **SEM image of FeNPs fabricated with *M. alternifolia* (TTO).** The scale bar represents 100 nm and a magnification of 120,000.

## Fourier-transform infrared spectroscopy

FTIR is an analytical method used to identify organic and inorganic materials that can acquire a spectrum of absorption of gas, liquid, and solid (*DeviC & Author, 2017*). Analysis spectra of the tea tree oil extract (Fig. 5) exhibited peaks at 1,096.88 and 1,044.44 cm$^{-1}$ for C-O bonds, and 1,379.12 cm$^{-1}$ vibration of the COOH (carboxylic acid) functional group (*Cárdenas-Alcaide et al., 2023*). Furthermore peaks were determined at 1,454.15 cm$^{-1}$ for O-H (hydroxyl) 1,645.87 cm$^{-1}$ for C = C (alkene), 1,966.76 cm$^{-1}$ for $N = C = S$ (isothiocyanate), 2,200.37 cm$^{-1}$ for C Ξ C (alkyne), 2,182.73 and 2,152.31 cm$^{-1}$ for S-C Ξ N (thiocyanate, 2,925.80 and 2,976.40 cm$^{-1}$ for C-H (alkane), 3,356.11 cm$^{-1}$ for N-H (primary amine), and 877.75 cm$^{-1}$ for C = C N-H (aliphatic primary amine). The IR spectra indicated aromatic biomolecules (1,645 87 cm$^{-1}$) and proteins (3,356.11 cm$^{-1}$) that could be the main ingredients in the investigated extract and might reduce Fe and Zn ions to NPs and be involved as capping and stabilizing agents. Proteins are known to be significant in reducing and stabilizing agents for NPs that prevent agglomeration (*El-Naggar et al., 2016*). The plant extract's components reduce Fe ions, and the water-soluble heterocyclic components are known to stabilize the NPs (*Kanagasubbulakshmi & Kadirvelu, 2017*).

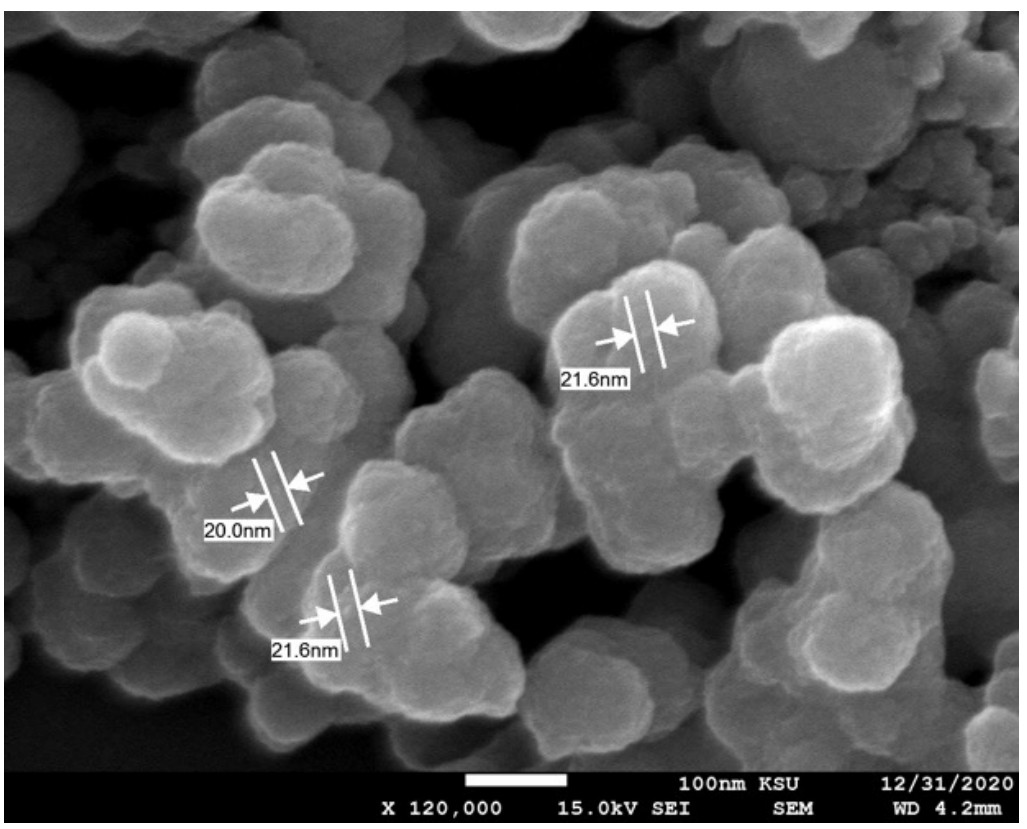

**Figure 4** **SEM image of ZnNPs fabricated with *M. alternifolia* (TTO).** The scale bar represents 100 nm and a magnification of 120,000.

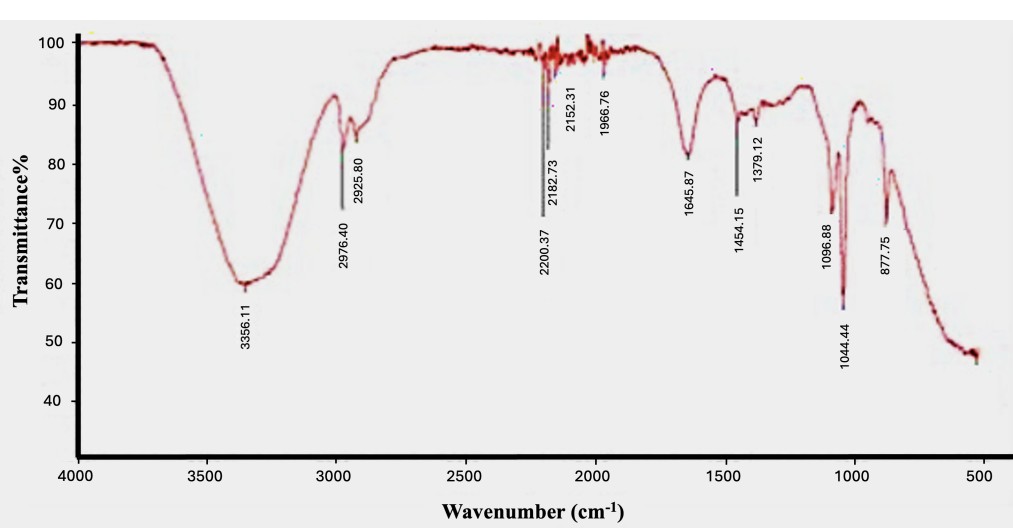

**Figure 5** **FT-IR image of the aqueous extract of TTO.**

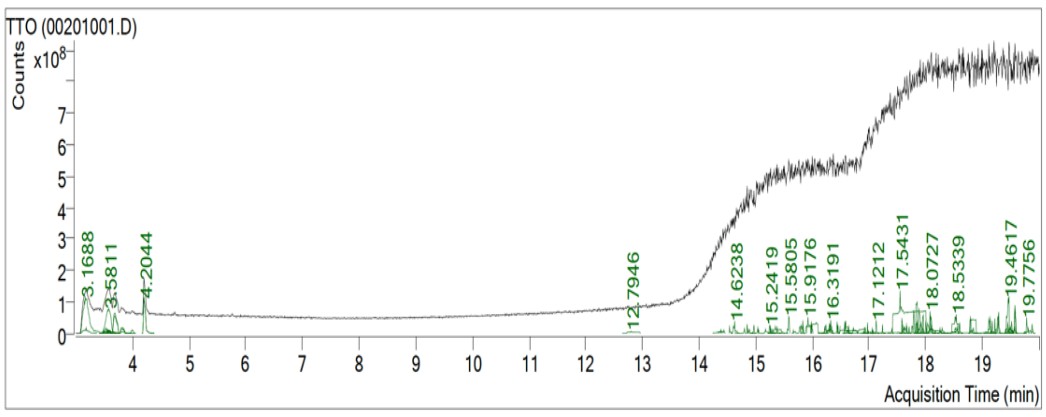

**Figure 6** Total ion chromatogram scan of the chemical compounds in TTO acquired from GC-MS analysis.

**Table 1** Selected chemical compounds detected from TTO by GC-MS analysis.

|   | Component RT | Compound name | Formula |
|---|---|---|---|
| 1 | 3.1688 | N,N-Bis (Carbobenzyloxy)-lysine methyl (ester) | $C_{23}H_{28}N_2O_6$ |
| 2 | 3.5811 | Benzene, 1-methyl-3-(1-methylethyl)- | $C_{10}H_{14}$ |
| 3 | 3.6877 | Gamma. –Terpinene | $C_{10}H_{16}$ |
| 4 | 3.6877 | Cyclohexene, 4-methylene-1 –(1- methylethyl)- | $C_{10}H_{16}$ |
| 5 | 3.8091 | 1,3-Cyclohexadiene,1-methyl-4- (1-Methylethyl)- | $C_{10}H_{16}$ |
| 6 | 4.2044 | Terpinen-4-ol | $C_{10}H_{18}O$ |
| 7 | 4.2044 | 3-Cyclohexen-1- ol, 4-methyl-1-(1-methylethyl)-, (R) - | $C_{10}H_{18}O$ |
| 8 | 19.8728 | Tetrakis (pentafluorobenzoyl)-hydrazine | $C_{28}F_{20}N_2O_4$ |

## Identification of phytochemicals through GC-MS analysis

In order to detect the chemical composition of the TTO and to find out the chemical compounds that could be responsible for its activity, GC-MS analysis was applied. Results indicated about 82 compounds with N,N-Bis (Carbobenzyloxy)-lysine methyl (ester), Benzene, 1-methyl-3-(1-methylethyl)-, Gamma–Terpinene, Cyclohexene, 4-methylene-1- (1-methylethyl)-, 1,3-Cyclohexadiene,1-methyl-4- (1-Methylethyl)-, Terpinen-4-ol, 3-Cyclohexen-1- ol, 4-methyl-1-(1-methylethyl)-, (R) and Tetrakis (pentafluorobenzoyl)-hydrazine as the major compounds (Fig. 6 and Table 1). Monoterpene hydrocarbons (Gamma-Terpinene) and alcohols (Terpinen-4-ol) were also detected in a previous study (*Kamel et al., 2022*). Our findings from the GC-MS and FTIR analysis could be well matched since Terpinen-4-ol and 3-Cyclohexen-1-ol, 4-methyl-1-(1-methylethyl)-, (R) beside N,N-Bis (Carbobenzyloxy)-lysine methyl (ester) and Tetrakis (pentafluorobenzoyl)-hydrazine were already suggested from the FTIR analysis due to the detected OH and amine groups respectively. The detected compounds could be responsible for the ability to reduce TTO and antimicrobial activity (*de Fátima Souto Maior et al., 2019*).

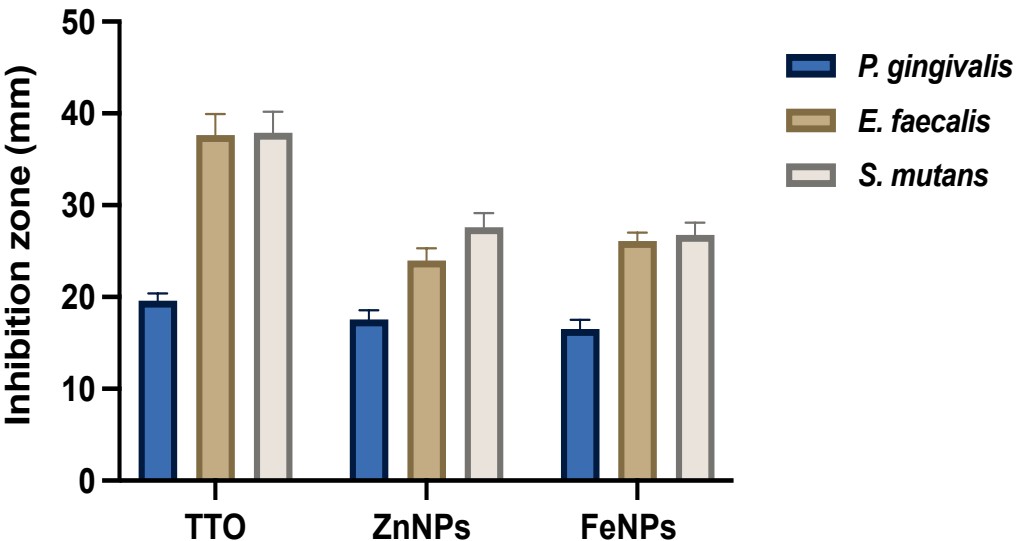

**Figure 7  Antibacterial activity of green synthesized TTO NPs against oral bacteria.** The effect was measured as a zone of growth inhibition presented in millimeters. Data are the mean $\pm$ SD ($n = 3$ replicates).

## Antimicrobial activity of extracts

Antibiotic resistance in bacteria and associated concerns pose a severe risk to human health; therefore, looking for substitutes is an important global priority (*Alqahtani, Othman & Mohammed, 2020*). The biosynthesis of NPs using plant extracts might solve the problem of antibiotic resistance caused by bacterial biofilm development (*Banerjee et al., 2020*) since many plant extracts have antibacterial capabilities and are biocompatibility, non-toxic, safe, and stable (*Xu et al., 2021*).

TTO has a long history of traditional applications by indigenous Australians (*Sharifi-Rad et al., 2017*). In the present study, TTO extract had potent antimicrobial activity against the tested oral bacterial species *P. gingivalis, E. faecalis,* and *S. mutans* with clear inhibition zones (19.58, 37.64, 37.89 mm) as presented in Fig. 7. The activity of the TTO against *E. faecalis* and *S. mutans* was significantly higher than that against *P. gingivalis*. There were no significant differences between *E. faecalis* and *S. mutans* in their response to TTO treatment. Surprisingly, ZnNPs and FeNPs prepared by TTO had a lower antibacterial effect against all tested strains compared to the TTO alone (Fig. 7). The activity of TTO could be linked to its active ingredients, detected by GC-MS analysis, such as terpinen-4-ol, which is one of the active components of TTO conferring antimicrobial activity to the oil (*Corona-Gómez et al., 2022*). The lower antibacterial activity of NPs compared to TTO could be related to the interaction mechanism of NPs when TTO was added to the metal salts for NP fabrication, which may lower their reactivity. The diversity of the chemical compounds detected in TTO could be the reason behind the reducing and capping ability of TTO; however, enhanced NP agglomeration might be the reason behind its low antibacterial effect. Clove oil could form silver NPs but were agglomerated (*Pervaiz et al., 2023*).

TTO has been shown to inhibit oral bacteria in root canals, effectively inhibit *E. faecalis,* and suppress its ability to form a biofilm (*Qi et al., 2021*), as well as having activity against *P. gingivalis* (*Mohammed et al., 2023a*). An earlier study identified the antimicrobial activity of TTO against Gram-negative and Gram-positive bacteria was caused due to disruption of membrane integrity, leading to an increase in the membrane permeability of bacteria, causing a loss of chemiosmotic control (*Cox et al., 2001*).

*P. gingivalis* was affected by TTO, ZnNPs, and FeNPs with inhibition zones of 19.58, 17.56, and 16.53 mm, respectively. *E. faecalis* was affected by TTO, ZnNPs, and FeNPs with inhibition zones of 37.64, 23.97, and 26.11 mm, respectively, as shown in Fig. 7. There were no significant differences between the activity of ZnNPs and FeNPs against *P. gingivalis* and *E. faecalis*. *S. mutans* were affected by TTO, ZnNPs, and FeNPs with inhibition zones of 37.89, 27.58, and 26.75 mm, respectively. Reports show the antibacterial activity of biogenic ZnONP against *S. aureus, E. coli*, and *S. Paratyphi* and FeNPs against *P. aeruginosa, E. coli, S. aure us*, and *B. subtilis* (*Theophil Anand et al., 2019*; *Vitta et al., 2020*). Furthermore, ZnONPs synthesized by *Acalypha fruticosa* leaf extracts exhibited antibacterial properties against *S. aureus, B. subtilis, P. vulgaris*, and *P. aeruginosa* (*Vijayakumar et al., 2019*). Since NPs have a high surface area for a greater interface of molecular interactions and smaller particles can penetrate bacterial cell walls and cause cell damage, enhanced antimicrobial effects were expected (*Din et al., 2015*; *Gudimalla et al., 2021*). However, we found in all cases, TTO alone was more active in the suppression of bacterial growth; this unexpected result is most probably attributed to agglomeration seen in the NPs or even the NPs concentration tested. Clumping of NPs may lessen the antimicrobial activity (*Mohammed et al., 2021*). Indeed, one of the drawbacks of synthesizing green metal NPs is the production of imperfect surface structures and heterogeneity in shape and phytochemicals involved in the reduction and capping process (*Bahrulolum et al., 2021*). Furthermore, a determinant for the stability of the green synthesized metal-NP is the concentration of the plant extract (*Sharma et al., 2022*); it could be the case that the concentration used in this study was not optimal to confer stability and maximum therapeutic efficacy. In addition, the phytochemicals attached around the central metal nucleus may have active functional groups that are not exposed outwardly to allow biological interactions to take place, hence limiting the therapeutic activity of the Green-NPs (*Saptarshi, Duschl & Lopata, 2013*). In support of this, we reported earlier that the functional groups might lend a role in repelling or attracting biological targets, which could be useful in precision medicine (*Mohammed et al., 2023b*), but concurrently, the lack of control over the specific phytochemicals and exposed functional groups around the surface of the NP may contribute to increasingly inactive NPs. Clumped NPs clearly would reduce biological reactivity due to the reduced surface area ratio, reducing the biological interaction interfaces. Optimal exposure of the NP surface allows for multiple ways of biological action, where RNA, DNA, and proteins can adsorb/interact, depending on hydrophobicity, hydrogen bonding, and ion-pair interactions at the nano-bio interface, causing conformational changes and loss of function (*Saptarshi, Duschl & Lopata, 2013*), resulting in altered signaling and lack of cellular integrity and function. Since the green NPs have active metabolites that would target various proteins, we would expect numerous modes of action

(*Corona-Gómez et al., 2022*; *Mohammed et al., 2023a*), but these modes of action are highly dependent on the factors discussed above. Several plant-active ingredients with antibacterial effects are known to stimulate ROS production in the cells, leading to cytotoxicity (*Li et al., 2021*), and this could be a mode of cytotoxic action of TTO. Further experiments designed to study the specific interactions of the NPs are needed to determine potential use as therapy.

## Tolerance determination

The MIC and MBC for the tested bacterial strains were examined using different concentrations of TTO. The MIC values and the MBC for TTO against all tested bacteria were 12.5 and 75 mg/mL. One particular study found the tolerance levels for all tested strains of bacteria using 100% undiluted TTO to be bactericidal against *E. coli*, *K. pneumonia*, and *S. aureus* (1.2, 1, and 5, respectively) when exposed to AgNPs synthesized using *Cinnamomum tamala* leaf extract (*Dash et al., 2020*).

## Potential antimicrobial synergy

Combining drugs can provide additive, synergistic, or antagonistic effects. Having an antagonist effect worsens therapeutic efficacy and can contribute to further bacterial resistance. To decipher the effects of the combination treatment of TTO with mouthwash, and standard antibiotics, metronidazole (flagyin) and amoxicillin, *P. Gingivalis* was treated with combination agents mixed at their lowest concentrations, and their activities were evaluated. *P. gingivalis* was sensitive to all treatments, as shown in Fig. 8. An incomplete additive antibacterial activity was seen when TTO was combined with standard antibiotics, flagyin, and amoxicillin. Hence indicating the feasibility of combining these antibiotics with TTO for improved therapeutic efficacy in treating periodontitis. Furthermore, an additive effect indicates that the modes of action of both combined active agents are distinct but since the maximal additive effect was not observed a partial block interaction may be occurring. Reports indicate a positive additive antimicrobial activity against *E.coil* with a combined mixture of *Cymbopogon khasianus* essential oil and streptomycin (*Langeveld, Veldhuizen & Burt, 2014*; *Singh & Katoch, 2020*). Another study found TTO and fluconazole mixture to be better at treating yeast infections than fluconazole alone (*Mertas et al., 2015*). Additive effects of antibiotics and essential oils arise due to distinct non-interacting mechanisms with targets affecting microbial growth. In this case, a reduction in the usage of antibiotics may be plausible. On the other hand, synergyism occurs when individual modes of action interact at certain pathways or targets which results in a positive enhanced activity for each. In cases of synergistic activity using far fewer concentrations of antibiotic may be possible.

On the other hand, TTO had antagonistic activity in *P. gingivalis* when combined with the mouthwash oral B and chlorhexidine. The combined activity of TTO with mouthwash and TTO and antibiotics against *E. faecalis* and *S. mutans* is presented in Figs. 9 and 10. Similar to *P. gingivalis,* TTO has an antagonistic effect on the mouthwash activity in *E. faecalis* and *S. mutans.* Flagyin is effective against anaerobic bacteria hence and, as expected, was not active against *E. faecalis* and *S. mutans* however combining flagyn with TTO partly antagonized the activity of TTO, which might be because of the dilution effect,

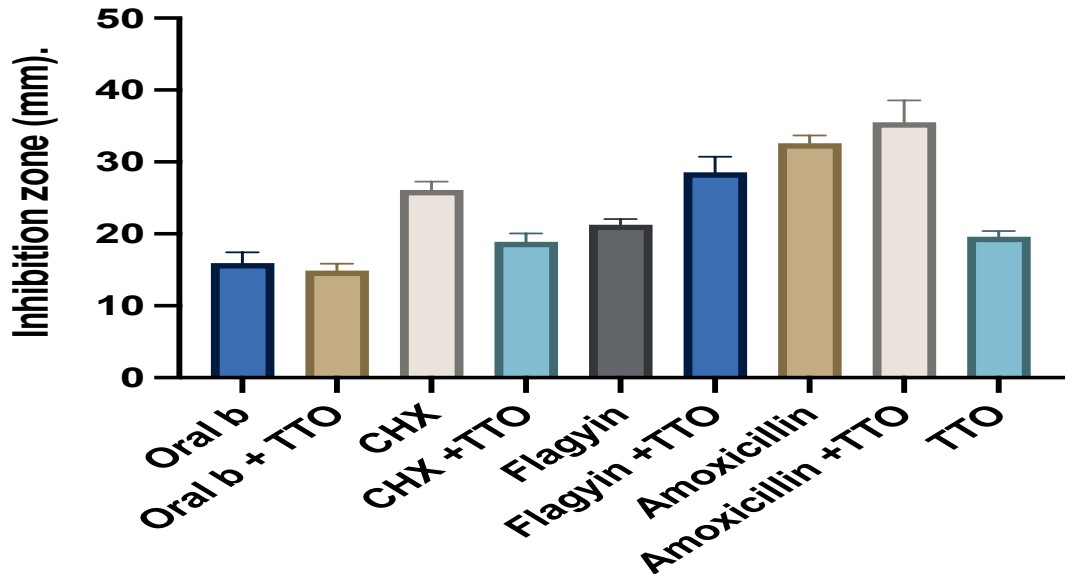

**Figure 8** **Effect TTO alone and combined with different antibiotics and mouthwash against *Porphyromonas gingivalis*.** The effect was measured as the growth inhibition zone in millimeters. Data are the mean ± SD ($n = 3$ replicates).

which was more notable in *E. faecalis*. TTO partly antagonized the activity of amoxicillin in both bacteria, indicating mechanistic interactions of an inhibitory nature to the normal targets of the tested agents. Similar findings were observed when TTO was combined with amphotericin B and tested against *Candia albicans* (*Van Vuuren, Suliman & Viljoen, 2009*). Such antagonistic effect indicates that the effect of TTO and antibiotic is varied according to the antibiotic and the microbes tested.

Differences in microbial responses to TTO alone or in combination with antimicrobial agents might be explained by variations in the structure of their cell membrane and individual targets. This study reflects the therapeutic relevance and clinical importance of using multiple antimicrobials or antibiotics simultaneously. Many patients may use TTO, natural agents, and off-the-counter mouthwashes without consultation with a doctor or pharmacist believing that there would be improved efficacy, not realizing the detrimental effects on their personal health and impact on microbial resistance.

## Morphological studies on treated bacteria

To study the antimicrobial mechanism of action of TTO and the NP counterparts, *P. gingivalis, E. faecalis,* and *S. mutans* were treated with TTO and the NPs for two hours and then visualized using SEM to determine differences in the bacterial morphology (Figs. 11, 12 and 13). TTO treatment caused a change in *P. gingivalis* cell size and decreased cell length compared to the control (Fig. 10). A previous report identified manuka oil to be antibacterial against *E. Coli* while shortening the size (*Alnaimat et al., 2015*).

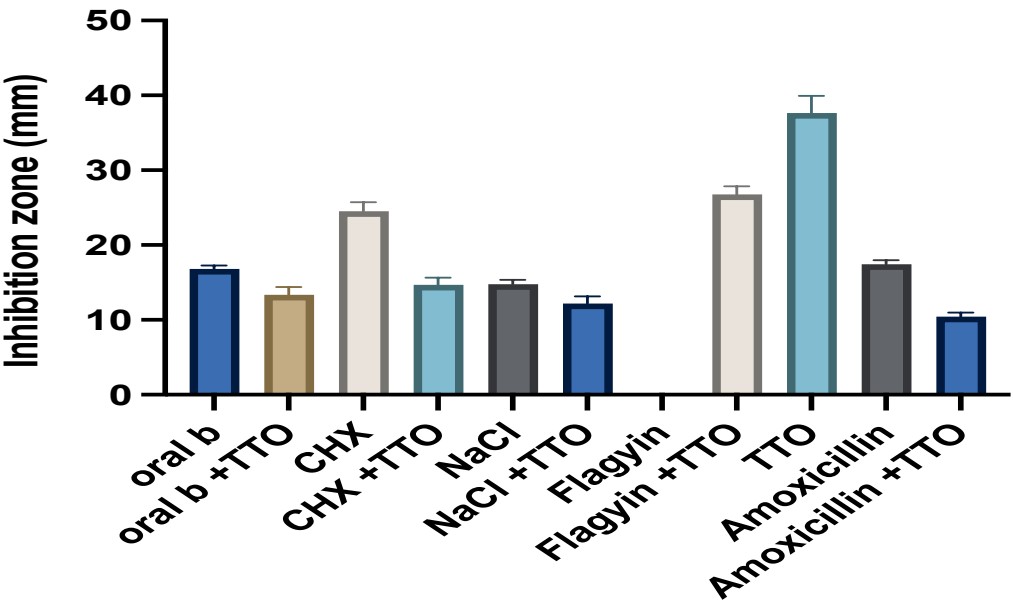

**Figure 9** **Effect of TTO alone and when combined with different antibiotics and mouthwash against** *Enterococcus faecalis.* The effect was measured as the zone of growth inhibition in millimeters. Data are the mean ± SD (*n* = 3 replicates).

Swelling of *E. faecalis* occurred on treatment with TTO and FeNP in relation to the untreated control, which could be caused by cell wall degradation allowing solution to penetrate inside the bacterial cell. On the other hand, TTO and ZnNPs treated *S. mutans* shrank in size compared to the control, likely caused by cell wall degradation leading to cellular components leaking outside the cell. Reports show various changes in the size and shape of bacteria after treatment with antimicrobial agents. Morphological changes in *Pseudomonas Orientalis* occur when treated with essential oils that lead to a coccoid shape (*Leja et al., 2019*). A recent study reported the morphological changes of *K. pneumoniae* on treatment with AgNPs prepared by soil fungal extracts (*Sonbol, Mohammed & Korany, 2022*). In addition, *Streptococcus pneumonia* and *Staphylococcus aureus* were treated with spherical AgNPs (5-30 nm) which were produced using *Conocarpus Lancifolius extract* resulting in cell wall disintegration and membrane rupture (*Oves et al., 2022*).

## CONCLUSION

TTO can be useful for metal-NP biosynthesis and has antibacterial efficacy against oral microbes. TTO extract alone was more efficacious than the Zn or FeNP counterparts. However, all tested agents had potential against tested microbes, leading to a change in oral microbe morphology. Our results go against many reports highlighting metal NP counterparts of herbal extracts as more efficacious than the herbal extract alone. However, this may be due to the clumping of the NPs or the concentration used in addition to the

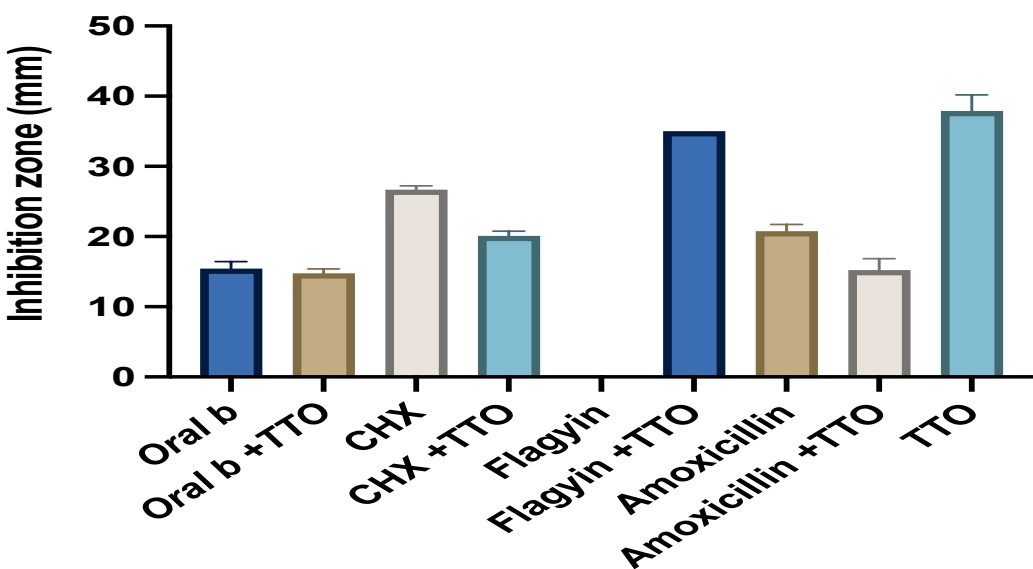

**Figure 10 Effect of TTO alone and when combined with different antibiotics and mouthwash against *Streptococcus mutans*.** The effect was measured as the zone of growth inhibition in millimeters. Data are the mean ± SD ($n = 3$ replicates).

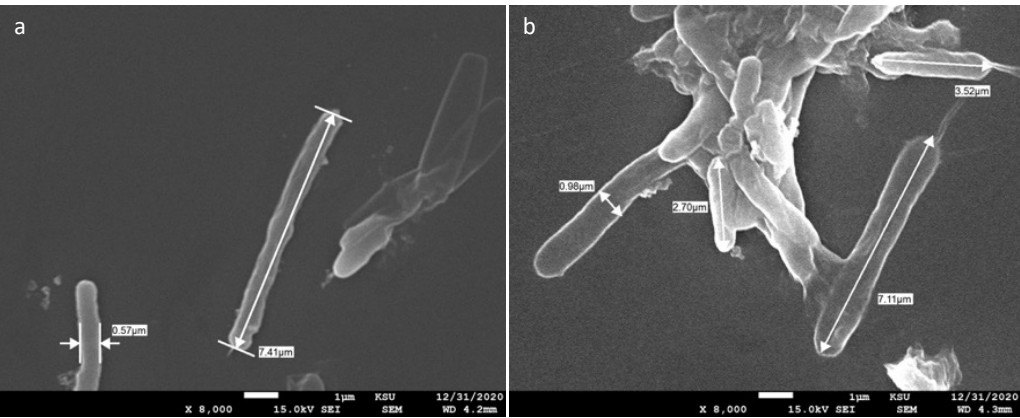

**Figure 11 SEM micrograph of *P. Gingivalis*.** Untreated (A) and treated with aqueous extract of TTO (B).

heterogeneity of phytochemicals in the capping process and limitations in the number of functional nano-bio interfaces. Furthermore, the additive effects of antibiotics and TTO or NPs may lead to new formulations to combat microbial resistance by using lower concentrations of antibiotics. The antagonistic effects seen with common mouthwashes represent a threat to therapeutic efficacy and heighten the risk of microbial resistance. *M. alternifolia* phytochemicals might be suitable for developing new antimicrobial drugs

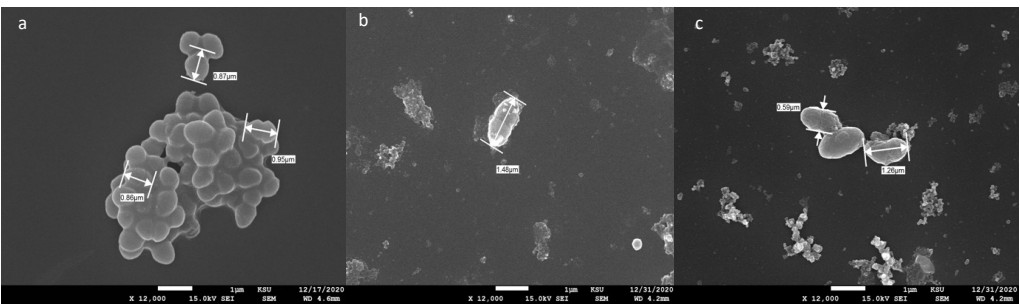

**Figure 12 SEM micrograph of *E. Faecalis*.** Untreated (A), treated with extract of TTO (B), and treated with phyto-fabricated FeNPs (C).

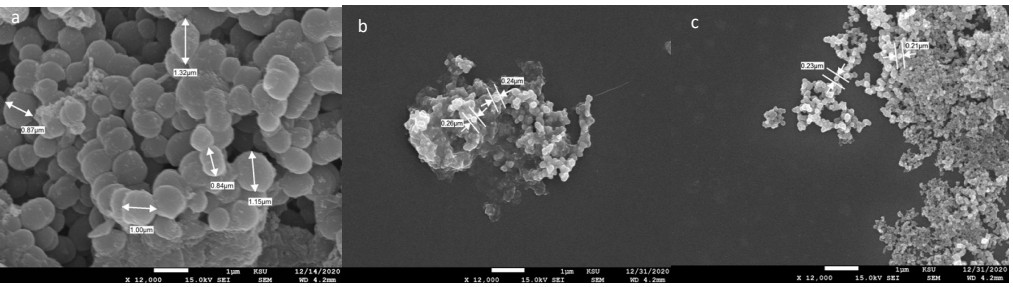

**Figure 13 SEM micrograph of *S. Mutans*.** Untreated (A), treated with extract of TTO (B), and treated with phyto-fabricated ZnNPs (C). The cells appear small in size compared to the untreated cells.

against oral microbes due to their noted efficiency as antimicrobial agents. In addition, we propose further experiments to optimise and regulate the the Green-NP synthesis process for maximum therapeutic efficacy.

## ACKNOWLEDGEMENTS

The authors are grateful to Kholoud Ali Baeshen for her technical assistance during the GC-MS experimental analysis.

### Funding

This work was funded by the Deanship of Scientific Research at Princess Nourah bint Abdulrahman University, through the Research Groups Program Grant no. (RGP-1443-0041). The funders had no role in study design, data collection and analysis, decision to publish, or preparation of the manuscript.

### Grant Disclosures

The following grant information was disclosed by the authors:

The Deanship of Scientific Research at Princess Nourah bint Abdulrahman University, through the Research Groups Program: RGP-1443-0041.

## Competing Interests

The authors declare there are no competing interests.

## Author Contributions

- Afrah E. Mohammed conceived and designed the experiments, performed the experiments, analyzed the data, prepared figures and/or tables, authored or reviewed drafts of the article, and approved the final draft.
- Reham M. Aldahasi performed the experiments, analyzed the data, prepared figures and/or tables, and approved the final draft.
- Ishrat Rahman conceived and designed the experiments, authored or reviewed drafts of the article, and approved the final draft.
- Ashwag Shami performed the experiments, prepared figures and/or tables, and approved the final draft.
- Modhi Alotaibi performed the experiments, prepared figures and/or tables, and approved the final draft.
- Munerah S. BinShabaib conceived and designed the experiments, authored or reviewed drafts of the article, and approved the final draft.
- Shatha S. ALHarthi conceived and designed the experiments, authored or reviewed drafts of the article, and approved the final draft.
- Kawther Aabed conceived and designed the experiments, performed the experiments, analyzed the data, prepared figures and/or tables, authored or reviewed drafts of the article, and approved the final draft.

## Data Availability

The raw data and graphs generated from the antimicrobial experiments is available in the Supplemental File.

## Supplemental Information

Supplemental information for this article can be found online at http://dx.doi.org/10.7717/peerj.17241#supplemental-information.

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
