# Peer review of "The antimicrobial activity of tea tree oil (Melaleuca alternifolia) and its metal nanoparticles in oral bacteria"

_PeerJ, doi:10.7717/peerj.17241_

## Round 0.1 · original submission · Major Revisions

Thank you for your submission. With careful attention to the reviewer comments, this manuscript will be suitable for publication at PeerJ. Please give special attention to reviewer comments focussed on improving figure readability and ensuring that the aspect ratios are not skewed. Reviewer 1 comments focussed on methodology and references to justify choices of concentration are especially important. The history of natural compounds for oral hygiene beyond Australia is also of interest (e.g. miswak in Africa, or maybe Neem in India?). Please be careful to use bacterial taxonomic names correctly (e.g. Streptococcus mutans, not capitalized on line 56). Finally a careful read through for grammar, readability and typos is essential (e.g. line 55, missing space in "disease(Dominy"

**Language Note:** The review process has identified that the English language must be improved. PeerJ can provide language editing services - please contact us at [email protected] for pricing (be sure to provide your manuscript number and title). Alternatively, you should make your own arrangements to improve the language quality and provide details in your response letter. – PeerJ Staff

·

Basic reporting

I hope this message finds you well. As a reviewer for the manuscript titled "The Antimicrobial Activity of Tea Tree Oil (Melaleuca alternifolia) and Its Metal Nanoparticles in Oral Bacteria," I am pleased to provide my evaluation and recommendations regarding the completion of the review process.

Upon thorough examination of the manuscript and considering the valuable insights from the author(s), I would like to offer the following perspectives:

1. The manuscript exhibits promising research potential; however, I suggest that the author(s) consider incorporating GC mass or HPLC analysis to elucidate the composition of the tea tree oil for enhanced clarity and scientific rigor.
2. It is essential to supplement the methodology section with references detailing the fabrication procedures for the Nanoparticles (NPs) employed in the study to reinforce methodological transparency.
3. To bolster the scholarly foundation of the research, I recommend including references following the statement pertaining to the determination of antibacterial activity using agar-well diffusion methods.
4. Clarity regarding the concentrations of the extract utilized for assessing antibacterial activity—whether it involved a single concentration or multiple concentrations—would significantly contribute to the study's credibility and reproducibility.
5. The inclusion of references supporting the method for selecting concentrations of tea tree oil, tea tree oil mouthwash (0.05%), and standard antibiotics used for evaluating antibacterial activity is crucial to fortify the research's scientific underpinning.
6. Within the "Results and Discussion" section, providing specifics, such as identifying the wavenumber indicative of aromatic biomolecules (lines 215-229), would enhance the manuscript's precision and scholarly merit.
7. The authors' observation regarding the lower antibacterial effect of ZnNPs and FeNPs prepared by TTO compared to TTO alone (Fig. 6) requires further explanation, necessitating elucidation to strengthen the study's conclusions.
8. Additionally, beyond the factors highlighted by the authors, I suggest considering other potential reasons for the unexpected results, such as altered properties, surface area, reactivity, and interaction mechanisms of nanoparticles, substantiated by pertinent citations for a comprehensive analysis.
9. Furthermore, it would be beneficial for the author(s) to outline the limitations of their study to provide a holistic understanding of the research scope and potential constraints.

In conclusion, I believe that addressing these suggestions would substantially augment the manuscript's scientific rigor and enhance its contribution to the field. I recommend providing the author(s) with the opportunity to revise the manuscript accordingly, ensuring the incorporation of these crucial aspects into the study.

Thank you for considering my input in the review process. I remain committed to supporting the enhancement of the manuscript's quality and scientific significance.

Warm regards,

The reviewer

Experimental design

No comment

Validity of the findings

Upon thorough evaluation of the research findings and methodologies employed in the study, I would like to commend the author(s) for their commendable efforts in investigating the antimicrobial activity of tea tree oil (TTO) and its metal nanoparticles (NPs) against oral bacteria. However, I would like to offer some insights into the validity and robustness of the presented findings.

The manuscript presents intriguing observations on the antimicrobial effects of TTO and its NPs against oral bacteria. However, to ensure the validity and reliability of these findings, several aspects warrant consideration:

1. Methodological Clarity: The manuscript would benefit from enhanced methodological clarity, particularly in detailing the fabrication procedures of the metal nanoparticles (NPs) derived from tea tree oil. A comprehensive description of the fabrication methods would strengthen the credibility of the study's findings.
2. Analytical Techniques: The incorporation of advanced analytical techniques, such as GC mass or HPLC analysis, could provide invaluable insights into the composition and characterization of the tea tree oil, thus bolstering the validity of the experimental outcomes.
3. Concentration Variability: The manuscript lacks explicit clarification on whether the antibacterial assays were conducted using a single concentration of the extract or involved multiple concentrations. Clarifying this aspect would contribute to the reproducibility and validity of the reported outcomes.
4. Referential Support: The inclusion of references supporting the rationale behind the selection of concentrations for tea tree oil, tea tree oil mouthwash (0.05%), and standard antibiotics for evaluating antibacterial activity would fortify the validity of the study's experimental design.
5. Interpretation of Results: Certain unexpected observations, such as the diminished antibacterial effect of NPs compared to TTO alone, necessitate further elucidation to ensure the validity and credibility of the findings. A comprehensive explanation, supported by relevant scientific literature, would strengthen the interpretation of these outcomes.

In conclusion, while the manuscript presents intriguing findings on the antimicrobial activity of tea tree oil and its NPs against oral bacteria, addressing the aforementioned aspects would significantly contribute to the validity and reliability

Additional comments

No comments

Reviewer 2 ·

Basic reporting

1-The manuscript requires strong linguistic review from a specialist in the field or a fluent English speaker.
2-The introduction shows a good understanding of the previous review, but it requires a lot of summarizing, arranging, and paraphrasing.
3-The provided figures need to improve their quality and resolution; figures legends must be rephrased more accurately on a scientific basis based on previously published references.
4- Material and methods part do not contain any scientific references,
please insert it according to section.
5- In results part, MBC results was mentioned where in methodology part was not mentioned at all, please explain.
6- Discussion Part needs to be improved.

Experimental design

The research within the scope of the journal.
Research question is clear and important to face the challenges of today, including the emergence of antibiotic-resistant bacterial species, and using ecofriendly resources instead.
Methods need to be improved and include scientific references.

Validity of the findings

no comment

Additional comments

-

---

## Round 0.2 · accepted · Accept

The reviewers have confirmed that all concerns have been addressed. Well done, thank you for your improvements. I am happy to say that the manuscript is ready for publication.

·

Basic reporting

No additional comments

Experimental design

No additional comments

Validity of the findings

The author(s) have improved the methodological clarity of the manuscript, especially regarding the fabrication procedures of the metal nanoparticles derived from tea tree oil. Clearly detailing these procedures is crucial for replicability and understanding.

Additional comments

No additional comments

Reviewer 2 ·

Basic reporting

All the required modification was done carefully by the authors

Experimental design

The Missing information was added

Validity of the findings

-

Additional comments

All the required modification was done by the authers